# Weakly Supervised Forgery Localization and Detection Method for Face Manipulations

## Abstract

The striking proficiency of generative models in producing and manipulating images with an unprecedented level of realism has elicited concerns regarding to malicious applications like face manipulation techniques. However, the majority of existing face forgery detection models are developed to provide only the real or fake binary label for a given image, which is not sufficient to identify the location of the manipulated area. In this paper, we propose a weakly supervised method for face manipulation localization and detection based on the Vision Transformer architecture, which only makes use of image-level labels and can realize forgery localization without extra pixel-level annotations. Unlike other weakly-supervised localization methods which conduct prediction directly depending on the feature of the single image, we design a novel weakly-supervised localization method that leverages statistical distribution characteristics of the entire dataset. It estimates multivariate Gaussian distributions for real and fake samples, and further uses the learned distributions to predict the location of the manipulated area. Additionally, based on the predicted mask, we propose a Distribution Centrality Learning to improve the compactness of patch embeddings around the distribution centers to further promote forgery localization. Additionally, we develop a new large-scale face manipulated image dataset, named DiffFMD, which is composed of various state-of-the-art diffusion-based generators and multiple sizes of facial manipulation regions.The experimental results demonstrate that the proposed method can achieve high detection and localization performance for face manipulation images.

## 1 Introduction

Deep generative models have made significant progress, including variational autoencoders (VAEs), generative adversarial models (GAN), computer graphics, and denoising diffusion probability models (DDPM). These models have achieved impressive results in image synthesis, image inpainting, and image denoising tasks. However, they have also been criticized for their potential to be used for malicious applications, such as face manipulation techniques (FaceSwap, 2019; DeepFakes, 2019; Thies et al., 2016; Wu et al., 2020), allowing for the creation of highly realistic facial forged images and videos with little human intervention. It poses a threat to security and privacy as these manipulated facial images may be abused for malicious purposes such as spreading fake news or falsifying evidence. Therefore, it is crucial to develop effective methods for detecting fake faces.

The majority of existing face forgery detection methods (Chollet, 2017; Wang et al., 2020; Qian et al., 2020; Zhao et al., 2021a) treat this task as a binary classification problem and provide only the 'real' or 'fake' binary label for a given image. However, it is insufficient to provide reliable evidence for recognizing the forged artifacts in the manipulated regions. There are two main challenges in the face forgery localization task: (i) pixel-level forged location annotations are non-trivial, and (ii) lack of evaluation benchmark for face forgery localization task. The deficiency of pixel-level annotations makes it unrealistic to deal with forgery localization problem in a fully supervised manner, thus, we devote to address forgery localization in a weakly supervised scenario, where only image-level annotations are available.

We investigate typical architectures for weakly-supervised forgery localization methods, as shown in Fig.1. The first category is the general explainability technique that uses gradient-weighted class activation maps (GradCAM) to highlight the most predictive regions for the "fake" label. The

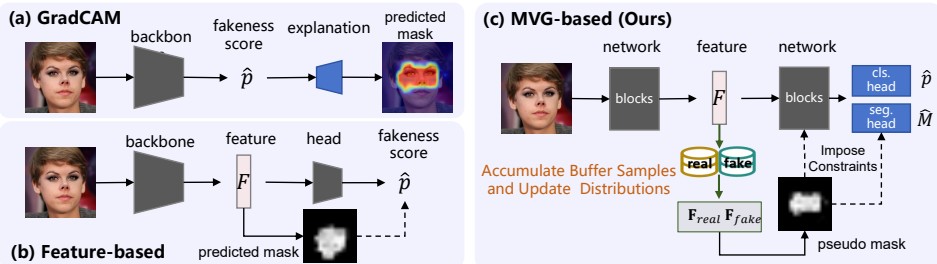

Figure 1: Compare with previous weakly-supervised forgery localization methods. (a) GradCAM produces visual explanations for decisions from forgery classifiers. (b) Feature-based methods (*e.g.* PatchForensics (Chai et al., 2020), Attention (Dang et al., 2020a)) use feature represetations from middle layers to produce forgery masks. (c) Ours predicts forgery masks based on multivariate Gaussian (MVG) distribution estimation, and leverages them to further promote network learning.

other feature-based methods, such as Patchforensics (Chai et al., 2020) and Attention (Dang et al., 2020a), leverage feature representations of middle layers to build the localization maps. However, these methods conduct forgery localization directly depending on the prediction or feature of the single input image. In contrast, we propose to preform forgery localization based on the statistical distribution characteristics of the entire dataset, inspired by the idea of unsupervised consistency learning of UIA-ViT (Zhuang et al., 2022). We predict the location by comparing the distances between the feature and the learned distributions. Thus, forgery localization prediction combines the learned distribution characteristics of vast real and fake samples, which can improve the reliability and accuracy of localization.

Specifically, the MVG-based forgery localization method leverages the middle patch embeddings of ViT to build multivariate Gaussian (MVG) distributions for real and fake samples. The sampling strategy is the most crucial process for the estimation of the distribution. Different from the fixed region sampling strategy of UIA-ViT (Zhuang et al., 2022), we innovatively design the adaptive sampling strategy, which can adjust adverse manipulated regions of different forged images. Based on the estimated real/fake MVG distributions, pseudo location annotations are generated by comparing the Mahalanobis distances between two distributions and patch embeddings from ViT's middle layer.

In addition, in order to further leverage the potential of predicted forgery mask for improving the detection and localization performance, we propose a Distribution Centrality Learning, which is a metric learning strategy aimed at enhancing the compactness of patch embeddings, which serves as an additional constraint for better forgery localization. DCL imposes a distance constraint between patch embeddings and their corresponding real/fake MVG distributions to guide them towards better compactness around the distribution centers.

Furthermore, we build a new dataset, named DiffFMD, which is composed of various state-of-the-art diffusion-based generators and multiple sizes of facial manipulation regions. The dataset is designed to establish the reliable evaluation benchmark for weakly-supervised face forgery localization and detection task in a realistic and challenging scenario. DiffFMD has three advantages: (1) face manipulation regions in multiple sizes. We generate random regions and sizes of manipulated masks for each facial image to emulate the manipulation regions that occur in real-world scenarios. (2) manipulation performed by several state-of-the-art diffusion-based generators. DiffFMD contains several state-of-the-art diffusion generative models, such as Repaint (Lugmayr et al., 2022), Paint-byExample (Yang et al., 2023), Stable Diffusion (Rombach et al., 2022). (3) a wide spectrum of source image datasets. We collect source facial images from two high-resolution image datasets, FFHQ (Kazemi & Sullivan, 2014) and CelebA (Liu et al., 2015), with an in-the-wild dataset Youtube selected from FF++ (Rossler et al., 2019), FFIW (Zhou et al., 2021) and CelebDF (Li et al., 2020b).

In summary, the main contributions of this paper are as follows:

- We develop a new weakly-supervised face forgery localization method, named MVG-FL, that leverages the evaluated Gaussian distributions to produce forgery maps. The innovative adaptive sampling strategy effectively helps iterative distribution estimation in the weakly-supervised scenario.

- We propose a distribution centrality learning strategy that utilizes the predicted location maps to enhance the compactness of patch embeddings. It serves as an additional constraint to further bolster the unsupervised forgery localization module.

- We develop a new dataset, named DiffFMD, that provides a wide spectrum of source image datasets and state-of-the-art diffusion-based generators, which is designed to evaluate the localization and detection performance of proposed method in a realistic and challenging scenario. Our method shows superior localization and detection performance on DiffFMD and widely-used FaceForensics++ dataset.

## 2 RELATED WORK

### 2.1 FACE FORGERY DETECTION AND LOCALIZATION

Early face manipulation techniques usually result in noticeable artifacts (Li et al., 2018; Yang et al., 2019; Matern et al., 2019) or inconsistencies in the generated face videos, which serve as important cues for early face forgery detection works. Besides seeking for visual artifacts, frequency clues have also been incorporated into forgery detection to enhance detection accuracy, such as Two-branch (Masi et al., 2020), F3-Net (Qian et al., 2020) and FDFL (Li et al., 2021). Meanwhile, attention mechanisms have proven effective in recent studies including Xception-Attn (Dang et al., 2020b) and Multi-attention (Zhao et al., 2021a) (Miao et al., 2021). Additionally, to address the challenge of generalizing to unseen forgeries, some works (Li & Lyu, 2018; Li et al., 2020a) focused on inevitable procedures in forgery, others observed that certain types of inconsistency exist across various kinds of forgery videos, including temporal inconsistency (Haliassos et al., 2021; Zheng et al., 2021; Guan et al., 2022), intra-frame inconsistency (Chen et al., 2021; Zhao et al., 2021b; Sun et al., 2021; Zhuang et al., 2022).

Contrasting with the flourishing of detection tasks, the face forgery localization has arguably received less attention (Songsri-in & Zafeiriou, 2019; Dang et al., 2020b; Kong et al., 2022). Moreover, most prior works address localization in a fully-supervised setting, which have access to image-level labels and pixel-level annotations. However, groundtruth manipulation masks might not always be available, especially for newly developed generators. There is a few works involving the weakly-supervised forgery localization study. WSCL (Zhai et al., 2023) exploits the multi-source consistency and inter-patch consistency, and promotes the capability to locate the manipulation regions in generic images. Dolos (Tânțaru et al., 2024) performs the weakly-supervised localization benchmark on their proposed dataset. But this work only modifies existing detection models to accommodate weakly-supervised training scenarios and achieves limited localization performance. To this end, this work tries to handle the difficulties towards the weak-supervised face manipulation localization task.

### 2.2 DIFFUSION GENERATED IMAGE DATASET

With the emergence of DMs and their increasing dominance, several works (Zhu et al., 2023; Lorenz et al., 2023; Wang et al., 2022; Sha et al., 2022) collect a dedicated dataset of pristine images and fake ones generated by several diffusion models. DiffusionDB (Wang et al., 2022) is one of the first large-scale text-to-image dataset, generated by Stable Diffusion. DEFAKE (Sha et al., 2022) follows two prompt-image datasets (Lin et al., 2014; Young et al., 2014) and conducts the generated image dataset based on four text-to-image generators (Rombach et al., 2022; Nichol et al., 2021; Ramesh et al., 2022). GenImage (Zhu et al., 2023) employs all real images from ImageNet, and generates images for each class by ADM (Dhariwal & Nichol, 2021), Stable Diffusion (Rombach et al., 2022), Wukong (Wuk), et al. ArtiFact (Rahman et al., 2023) collects a large range of image categories, and its synthetic dataset consists of multiple GANs, DMs and other miscellaneous generators. Most datasets are built upon natural scene images and only cover text-to-image diffusion generation methods. They leave out image-to-image inpainting generators and represent only a partial coverage of diffusion-generated images which is relatively less challenging for detection. A recent work Dolos (Tânțaru et al., 2024) proposes a locally and fully manipulated image dataset. But Dolos only covers a few of early diffusion-based generators, and the size of dataset is small. Therefore, our work focus on identifying locally edited facial images manipulated by image-to-image inpainting diffusion models. We aim to make a valuable contribution to the research community by addressing a critical gap in synthesis image detection and localization.

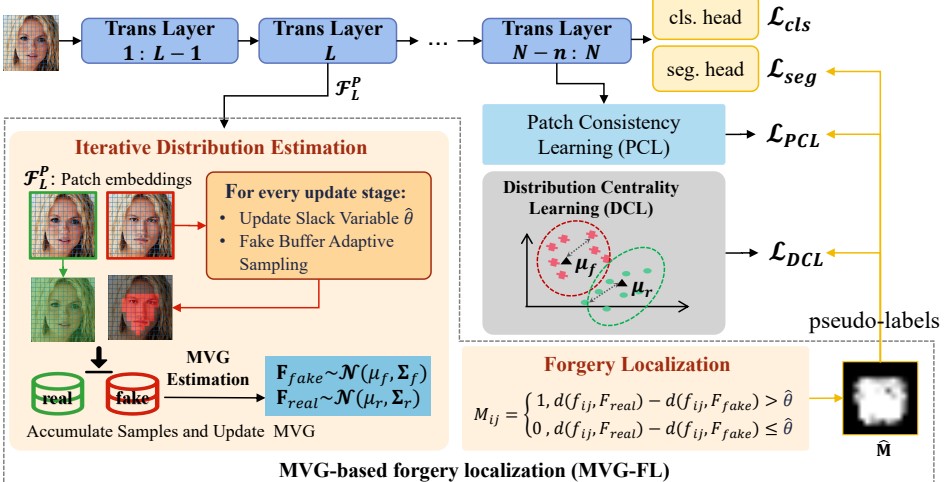

Figure 2: An overview of the proposed method. Patch embeddings $\mathcal{F}_L^P$ from layer $L$ are accumulated to estimate approximate forgery locations in MVG-FL module. The pseudo forgery mask label from localization module are then feed into the Distribution Centrality Learning (DCL) and Patch Consistency Learning (PCL) to harness the compactness of patch embeddings and guide consistency learning. The segmentation head is trained under the supervision of the pseudo labels.

## 3 METHOD

### 3.1 OVERVIEW

We use the UIA-ViT (Zhuang et al., 2022) as our baseline model. As shown in Fig.2, the patch embeddings $\mathcal{F}_L^P$ from layer $L$ are accumulated to estimate approximate forgery locations in the MVG-FL module. The predicted location maps from the forgery localization module are then fed into the Distribution Centrality Learning (DCL) and Patch Consistency Learning (PCL) to harness the compactness of patch embeddings and guide consistency learning. The PCL module is followed the design of UIA-ViT. The segmentation head is trained under the supervision of the pseudo labels.

### 3.2 MVG-BASED FORGERY LOCALIZATION (MVG-FL)

The main idea of the weakly supervised forgery localization is to leverage multivariate Gaussian (MVG) distribution to represent the real/fake patch features from ViT's middle layer, and then generate the predicted location map by comparing the distances between the patch embeddings and two estimated distributions. MVG distributions are iteratively updated during training, through following distribution estimation method.

#### 3.2.1 MVG DISTRIBUTION ESTIMATION

The mathematical model of multivariate Gaussian distribution can be represented as $\mathcal{N}(\boldsymbol{x}; \boldsymbol{\mu}, \boldsymbol{\Sigma})$, where $\boldsymbol{\mu}$ and $\boldsymbol{\Sigma}$ represent the mean vector and the symmetric covariance matrix. Let $\boldsymbol{x}_{real}$ denote the patch feature extracted from layer $L$ of a real sample, and $\boldsymbol{x}_{fake}$ represent the patch feature extracted from layer $L$ within the forgery region of a fake sample. We model the probability density function (PDF) of $\boldsymbol{x}_{real}$ using MVG, defined as:

$$F_{real}(\boldsymbol{x}_{real}) = \frac{1}{\sqrt{(2\pi)^D \|det\boldsymbol{\Sigma}_r\|}} e^{-\frac{1}{2}(\boldsymbol{x}_{real}-\boldsymbol{\mu}_r)^T \boldsymbol{\Sigma}_r^{-1}(\boldsymbol{x}_{real}-\boldsymbol{\mu}_r)}, \tag{1}$$

where $\boldsymbol{\mu}_r \in \mathbb{R}^D$ and $\boldsymbol{\Sigma}_r \in \mathbb{R}^{D \times D}$ represent the mean vector and the symmetric covariance matrix of the real distribution, respectively. Similarly, the PDF of $\boldsymbol{x}_{fake}$ is defined as $F_{fake}(\boldsymbol{x}_{fake})$ with mean vector $\boldsymbol{\mu}_f$ and covariance matrix $\boldsymbol{\Sigma}_f$.

During the training process, $F_{real}$ and $F_{fake}$ are updated based on new $(\boldsymbol{\mu}_r, \boldsymbol{\Sigma}_r)$ and $(\boldsymbol{\mu}_f, \boldsymbol{\Sigma}_f)$, which are approximated using the sample mean and sample covariance calculated from observations

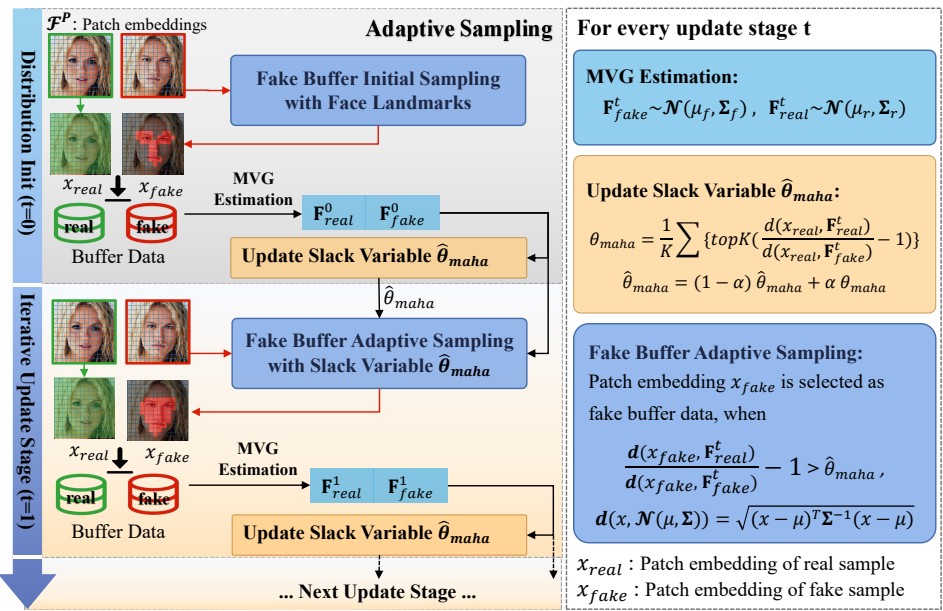

Figure 3: The pipeline of the distribution estimation with the sampling strategy: Adaptive Sampling.

$(\boldsymbol{x}_{real}^1, \boldsymbol{x}_{real}^2, ..., \boldsymbol{x}_{real}^n \in \mathbb{R}^D)$ and $(\boldsymbol{x}_{fake}^1, \boldsymbol{x}_{fake}^2, ..., \boldsymbol{x}_{fake}^n \in \mathbb{R}^D)$. We accumulate feature observations from each mini-batch of training samples, and update two MVG distributions experimentally every epoch.

### 3.2.2 ADAPTIVE SAMPLING

In order to achieve accurate distribution estimation, it is crucial to develop a sound sampling strategy for observation selection. We introduce an adaptive sampling strategy that adaptively adjusts the sampling region during iterative updates.

To be concrete, the observations selection of fake buffer is based on the relative distances between patch features and previously estimated distributions, which are regulated by a slack variable. The slack variable is the decision threshold of relative distances and is updated using the Exponential Moving Average (EMA) at each update stage. The total pipeline is shown in Fig.3.

At the initial training epoch, the distributions $F_{real}$ and $F_{fake}$ are estimated using the sample mean and covariance from patch features of real image and forged image, respectively.(Samples of the initial fake buffer are selected from face landmark region.) However, this initialization for $F_{fake}$ is only approximate due to the presence of both real and manipulated patches within a forged image. Note that in the ideal scenario, the feature of a real image patch is more closely related to $F_{real}$ than $F_{fake}$, whereas for a manipulated patch, it is the opposite. Thus, the relative distances between the patch features and the estimated distribution $F_{real}, F_{fake}$ can be utilized to select the feature observations for updating the fake buffer.

However, during early training stages, the feature extractor is less informative and the distribution estimation is not accurate enough. Simply comparing the distances would lead to lots of deviation samples in the fake buffer, and then have an undesirable impact on the distribution estimation. Then we think about the feasibility to regulate the decision threshold of relative distances. Based on the prior knowledge, the feature of a real image patch is more closely related to $F_{real}$ than $F_{fake}$, then the decision threshold of relative distances should coordinate with this circumstance. Therefore, we incorporate prior knowledge into introducing the slack variable $\theta_{maha}$, defined as the maximum

relative distance between patch features of real samples $\boldsymbol{x}_{real}$ and two distributions $F_{real}, F_{fake}$:

$$\theta_{maha} = \frac{1}{K} \sum topK(\frac{d(\boldsymbol{x}_{real}, F_{real})}{d(\boldsymbol{x}_{real}, F_{fake})} - 1), \tag{2}$$

$$\hat{\theta}_{maha} = (1 - \alpha)\hat{\theta}_{maha} + \alpha\theta_{maha}, \tag{3}$$

where $\alpha$ is the moving parameter of Exponential Moving Average. The utilization of EMA can prevent instability issues that may arise from early MVG distribution during the update stage. Here, we adopt Mahalanobis distance as the measure for calculating the distance between patch feature $x$ and the MVG distributions, defined as:

$$d(\boldsymbol{x}, F_{real}) = \sqrt{(\boldsymbol{x} - \boldsymbol{\mu}_r)^T \boldsymbol{\Sigma}_r^{-1}(\boldsymbol{x} - \boldsymbol{\mu}_r)}, d(\boldsymbol{x}, F_{fake}) = \sqrt{(\boldsymbol{x} - \boldsymbol{\mu}_f)^T \boldsymbol{\Sigma}_f^{-1}(\boldsymbol{x} - \boldsymbol{\mu}_f)}, \tag{4}$$

where $(\boldsymbol{\mu}_r, \boldsymbol{\Sigma}_r)$ and $(\boldsymbol{\mu}_f, \boldsymbol{\Sigma}_f)$ are the mean and covariance of the MVG distributions $F_{real}, F_{fake}$.

The $\hat{\theta}_{maha}$ serves as an adjustable threshold of relative distances for discriminating between real and fake patches. When the patch features meet the decision condition controlled by the slack variable $\hat{\theta}_{maha}$, they are selected to the next fake buffer and treated as feature observations for updating fake distribution. The decision condition is formulated as follows:

$$\frac{d(\boldsymbol{x}_{fake}, F_{real})}{d(\boldsymbol{x}_{fake}, F_{fake})} - 1 > \hat{\theta}_{maha}. \tag{5}$$

After accumulating feature observations every iterative training epoch, $F_{real}$ and $F_{fake}$ are updated with new estimates $(\boldsymbol{\mu}_r, \boldsymbol{\Sigma}_r)$ and $(\boldsymbol{\mu}_f, \boldsymbol{\Sigma}_f)$.

Given MVG distributions of real and fake, the distances between the patch embeddings $\mathcal{F}_P$ from layer $L$ and MVG distributions are used for forgery location prediction. Assume $f_{ij} \in \mathbb{R}^D$ is the particular feature in position $(i, j)$ of $\mathcal{F}_P$. For fake samples, the predicted location map $\mathbf{M} \in \mathbb{R}^{P \times P}$ is defined as a binary distance comparison map, where $P^2$ is the number of patch embeddings. The annotation is predicted as 0 when the patch feature is more closed to $F_{real}$ than $F_{fake}$, and otherwise predicted as 1, formalized as:

$$\mathbf{M}_{ij} = \begin{cases} 1, d(f_{ij}, F_{real}) - d(f_{ij}, F_{fake}) > \hat{\theta}_{maha} \\ 0, d(f_{ij}, F_{real}) - d(f_{ij}, F_{fake}) \leqslant \hat{\theta}_{maha} \end{cases}. \tag{6}$$

Note that location map $\mathbf{M}$ is fixes as the all-zero matrix for real samples.

### 3.3 DISTRIBUTION CENTRALITY LEARNING (DCL)

In the weakly-supervised localization module, the predicted location map from the forgery location module is only dependent on the estimated MVG distributions, which lacks other specific constraint. We further design a metric learning strategy to directly constrain the distance relationship between the predicted location map and the MVG distributions. It contributes to boost the compactness of patch embeddings, and guide them more compact with the centers of their predicted real/fake distributions. In details, with the help of the unsupervised forgery localization, the predicted location map indicates the real/fake attribute of every patch embeddings. Then DCL conducts a distance learning to make fake patch embeddings close to the distribution mean of $F_{fake}$, as well as make real patch embeddings close to the distribution mean of $F_{real}$. The distribution centrality loss is defined as:

$$\mathcal{L}_{DCL} = \sum_{i,j\{M_{ij}=1\}} (\boldsymbol{x}_f^{ij} - \boldsymbol{\mu}_f)^2 + \sum_{i,j\{M_{ij}=0\}} (\boldsymbol{x}_f^{ij} - \boldsymbol{\mu}_r)^2 + \sum_{i,j} (\boldsymbol{x}_r^{ij} - \boldsymbol{\mu}_r)^2, \tag{7}$$

where $\boldsymbol{\mu}_r$ and $\boldsymbol{\mu}_f$ are mean vectors of MVG distributions $F_{real}$ and $F_{fake}$. In the given image, the approximate location map $M$ from forgery localization module indicates the location of fake patch (where $M_{ij} = 1$) and real patch (where $M_{ij} = 0$). Then DCL utilizes the Euclidean distance to represent the distance relationship between fake/real patch embeddings and the distribution means of $F_{fake}, F_{real}$, which is optimized to decease during training. It can benefit to improve the compactness of patch embeddings and realize better forgery localization.

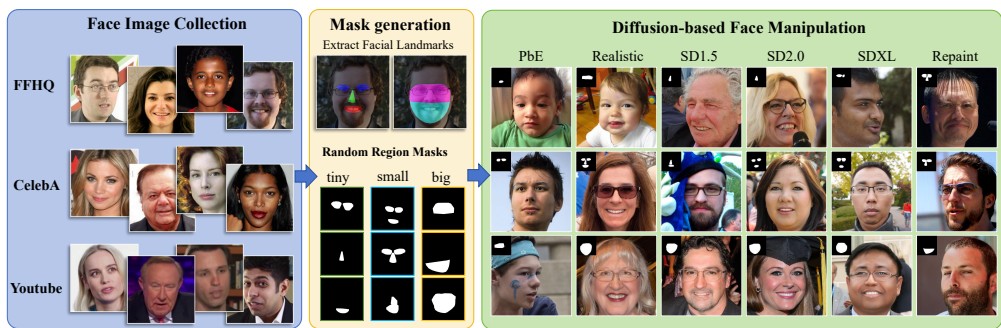

Figure 4: The DiffFMD dataset is composed based on three face image datasets. Three random mask regions are generated. Six diffusion-based generators are used to produce facial manipulation results.

### 3.4 MULTIPLE TRAINING OBJECTIVES

The binary cross entropy loss ($\mathcal{L}_{cls}$, $\mathcal{L}_{seg}$) is used to train the classification head and segmentation head. We also use the patch consistency learning ($\mathcal{L}_{PCL}$) on middle-layer attention maps to constraint the relationship between different patch embeddings, following the UIA-ViT (Zhuang et al., 2022). The total loss functions of the proposed method are described as:

$$\mathcal{L}_{total} = \mathcal{L}_{cls} + \lambda_1 \mathcal{L}_{PCL} + \lambda_2 \mathcal{L}_{DCL} + \lambda_3 \mathcal{L}_{seg}, \tag{8}$$

where $\lambda_1$, $\lambda_2$, $\lambda_3$ are hyper-parameters used to balance different losses.

## 4 EXPERIMENT

### 4.1 IMPLEMENTATION DETAILS

We adopt ViT-Base (Dosovitskiy, 2020) as backbone. In the MVG-FL method, the top-K parameter K in Eq.2 is set to 48. and the moving parameter $\alpha$ in Eq.3 is set to 0.95. Additionally, the patch embeddings from layer 6 are utilized for distribution estimation, and the average attention map from layer 8 to layer 12 are used for patch consistency learning (*i.e.*, $L = 6, n = 4, N = 12$). In the loss function of Sec.3.4, we set the weight $\lambda_1$, $\lambda_2$, $\lambda_3$ to 0.01, 0.001, 0.2.

### 4.2 DIFFFMD CONSTRUCTION

As shown in Figure 4, DiffFMD dataset is composed based on three real face image datasets, FFHQ (Kazemi & Sullivan, 2014), CelebA (Liu et al., 2015), Youtube (Rossler et al., 2019; Li et al., 2020b; Zhou et al., 2021). We have pre-defined three size levels of manipulation regions, categorized as tiny, small and big. The tiny is assigned as one of the eyes, nose, or mouth regions. The small is the random combination of eyes, nose, and mouth. Lastly, the big is one of the upper half face, the bottom half face, or the entire face. We use SOTA diffusion-based inpainting generators for generating manipulated face imags: PaintbyExample (Yang et al., 2023), Repaint (Lugmayr et al., 2022), Stable Diffusion (Rombach et al., 2022), and Realistic Vision. The dataset is composed of 272k images for training and 7k images for testing. More details are provided in the Appendix.

### 4.3 EXPERIMENTS ON DIFFFMD

We first evaluate the localization and detection performance on DiffFMD dataset. Models are developed on the training datasets (PaintbyExample, Realistic and SDXL), and then evaluated on all test datasets consisted of six generators (PaintbyExample, Realistic, SDXL, SD1.5, SD2.0, Repaint). Additionally, a recent work (Tânțaru et al., 2024) released a facial manipulation dataset called Dolos, generated by a few of diffusion-based and GAN-based generators. We utilize the inpainting test dataset of this work as the extra test dataset for our experiments, represented as 'Dolos' in Table 1.

In Table 1, we report the pixel-level AUC, pixel-level F1-score and image-level AUC on the DiffFMD dataset and the inpainted (Tânțaru et al., 2024) dataset. GradCAM (Selvaraju et al., 2017) is a traditional baseline method in the weakly-supervised localization task. Here we endow the ViT network with localization capabilities by applying GradCAM on the activations produced by block 8. Xception-Attn (Dang et al., 2020a) is the augmented version of Xception backbone with a learned

Table 1: Experimental results on DiffFMD datasets.The training datasets are underlined.

| Methods | Metrics | Test Datasets | | | | | | | | |
|---|---|---|---|---|---|---|---|---|---|---|
| | | PbE | Realistic | SDXL | SD1.5 | SD2.0 | Repaint | Dolos | FF++ | Avg |
| GradCAM (Selvaraju et al., 2017) | pix-AUC | 86.95 | 76.89 | 85.02 | 71.24 | 63.98 | 45.45 | 50.59 | 47.85 | 66.00 |
| | pix-F1 | 50.23 | 45.39 | 49.54 | 30.29 | 21.65 | 3.56 | 6.35 | 5.85 | 26.61 |
| | img-AUC | 95.99 | 97.73 | 93.28 | 83.19 | 77.60 | 58.29 | 60.09 | 60.69 | 78.36 |
| Xception-Attn (Dang et al., 2020a) | pix-AUC | 84.61 | 85.57 | 84.39 | 85.85 | 80.90 | 66.55 | 58.87 | 68.46 | 76.90 |
| | pix-F1 | 50.85 | 50.20 | 48.23 | 48.84 | 26.19 | 3.11 | 2.03 | 1.09 | 28.82 |
| | img-AUC | 96.46 | 97.56 | 95.82 | **87.74** | 80.09 | 54.97 | 60.20 | 56.80 | 78.71 |
| Patchforensics (Chai et al., 2020) | pix-AUC | 89.60 | 86.29 | 90.35 | 90.64 | 70.86 | 65.54 | 52.76 | 47.83 | 74.23 |
| | pix-F1 | 39.92 | 39.94 | 40.22 | 32.11 | 25.66 | 24.82 | 27.87 | 42.14 | 34.08 |
| | img-AUC | **97.05** | 96.95 | 92.94 | 84.81 | 78.15 | 58.17 | 59.58 | 50.64 | 77.29 |
| WSCL (Zhai et al., 2023) | pix-AUC | 75.35 | 79.58 | 78.85 | 78.32 | 79.16 | 49.71 | 56.50 | 61.21 | 69.83 |
| | pix-F1 | 32.58 | 31.25 | 32.83 | 29.12 | 31.77 | 10.96 | 9.11 | 3.50 | 22.64 |
| | img-AUC | 94.12 | 97.97 | **96.34** | 87.39 | 80.10 | 49.56 | 60.81 | 54.56 | 77.61 |
| POT (Wang et al., 2025) | pix-AUC | 89.90 | 89.89 | 88.28 | 87.13 | 77.41 | 49.14 | 57.57 | 62.61 | 75.24 |
| | pix-F1 | 59.39 | **50.59** | 48.94 | 48.29 | 35.60 | 5.46 | 10.65 | 8.04 | 33.37 |
| | img-AUC | 95.78 | 95.90 | 93.46 | 83.88 | 78.37 | 57.57 | 59.91 | 52.65 | 77.19 |
| UIA-ViT (Zhuang et al., 2022) | pix-AUC | 86.88 | 88.89 | 87.89 | 90.55 | 89.77 | 86.29 | 69.16 | 83.59 | 85.38 |
| | pix-F1 | 35.01 | 34.24 | 34.52 | 24.76 | 25.25 | 26.73 | 39.07 | 56.28 | 34.48 |
| | img-AUC | 96.04 | 97.95 | 93.15 | 84.00 | 77.12 | 59.22 | 60.08 | 61.66 | 78.65 |
| Ours | pix-AUC | **91.78** | **93.68** | **92.23** | **93.56** | **92.86** | **88.51** | **69.32** | **84.61** | **88.32** |
| | pix-F1 | **51.05** | 47.84 | **50.48** | **51.11** | **50.93** | **45.76** | **39.85** | **63.58** | **50.08** |
| | img-AUC | 96.31 | **98.13** | 94.34 | 85.43 | **81.55** | **60.48** | **60.91** | 62.68 | **79.98** |

Table 2: Experimental results on FF++(c23) dataset. Models are trained on five manipulations.

| Methods | Metrics | Test Datasets | | | | | |
|---|---|---|---|---|---|---|---|
| | | Deepfake | Face2Face | FaceSwap | NeuralTexture | FaceShifter | Average |
| GradCAM (Selvaraju et al., 2017) | pix-AUC | 36.84 | 43.33 | 44.85 | 44.51 | 50.64 | 44.03 |
| | pix-F1 | 5.58 | 10.55 | 9.84 | 10.40 | 16.22 | 10.52 |
| | img-AUC | 96.35 | 95.20 | 94.05 | 92.29 | 96.58 | 94.89 |
| Xception-Attn (Dang et al., 2020a) | pix-AUC | 80.50 | 64.66 | 84.34 | 49.67 | 82.62 | 72.36 |
| | pix-F1 | 10.59 | 10.52 | 20.11 | 2.45 | 14.56 | 11.65 |
| | img-AUC | 94.20 | 96.40 | 96.67 | 90.62 | 95.74 | 94.73 |
| Patchforensics (Chai et al., 2020) | pix-AUC | 73.54 | 58.31 | 53.64 | 57.02 | 60.61 | 60.62 |
| | pix-F1 | 47.24 | 58.40 | 35.71 | 55.90 | 53.69 | 50.19 |
| | img-AUC | 87.17 | 82.32 | 75.17 | 85.98 | 89.44 | 84.02 |
| WSCL (Zhai et al., 2023) | pix-AUC | 93.06 | 78.52 | 78.67 | 70.71 | 85.81 | 81.35 |
| | pix-F1 | 60.60 | 73.73 | 54.92 | 64.92 | 74.31 | 65.70 |
| | img-AUC | 93.78 | 93.24 | 81.41 | 90.15 | 90.39 | 89.79 |
| POT (Wang et al., 2025) | pix-AUC | 82.87 | 75.62 | 87.31 | 65.39 | 84.36 | 79.11 |
| | pix-F1 | 70.54 | 73.67 | 66.83 | 60.92 | 77.72 | 69.94 |
| | img-AUC | 92.49 | 94.32 | 90.31 | 91.45 | 93.32 | 92.38 |
| UIA-ViT (Zhuang et al., 2022) | pix-AUC | 95.58 | 80.98 | 94.60 | **68.59** | 90.88 | 86.13 |
| | pix-F1 | 63.58 | **74.86** | 63.51 | 64.17 | 78.24 | 68.87 |
| | img-AUC | 96.69 | 95.58 | **96.30** | 90.72 | 96.86 | 95.23 |
| **Ours** | pix-AUC | **96.48** | **81.22** | **96.72** | 66.79 | **92.52** | **86.75** |
| | pix-F1 | **76.21** | 74.07 | **70.96** | **64.88** | **80.05** | **73.23** |
| | img-AUC | **97.64** | **95.62** | 95.51 | **93.80** | **97.18** | **95.95** |

attention mask that is used to modulate the feature maps. Patchforensics (Chai et al., 2020) is a truncated image classification network, which takes the feature activations after a few layers and projects them to a patch-level score.WSCL (Zhai et al., 2023) is a weakly-supervised localization method towards generic image manipulation detection task. POT(Wang et al., 2025) is a recently weakly supervised semantic segmentation method, which enhances CAM predictions by dividing features into multiple clusters and activating each cluster using its prototype. UIA (Zhuang et al., 2022) can predict the manipulated region using the learned MVG distributions and middle features from ViT architecture. All models are reproduced on the DiffFMD dataset, and then evaluate their localization and detection performances quantitatively on seven test datasets.

Table 1 shows that our method achieves outstanding localization performance compared to the other methods, including in-domain scenario (PaintbyExample, Realistic and SDXL) and cross-domain

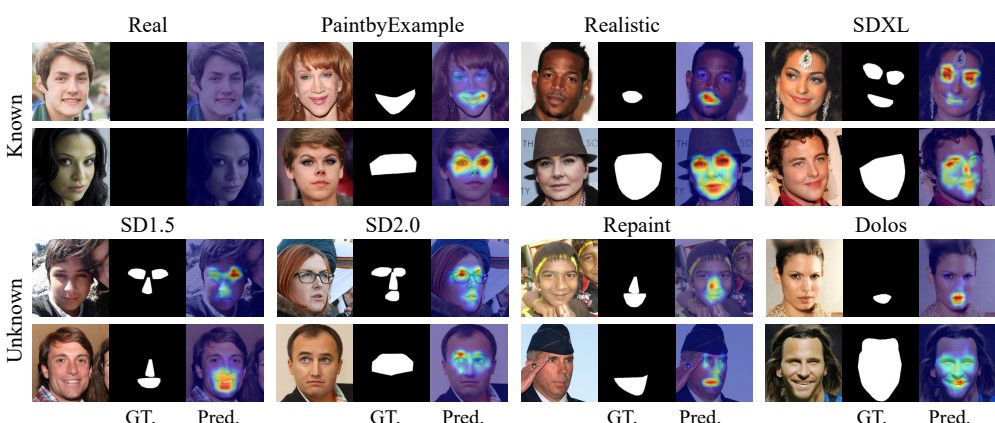

Figure 5: Predicted Location Map of DiffFMD test datasets.

scenario (SD1.5, SD2.0, Repaint, inpainted (Tânțaru et al., 2024)). It demonstrates the effectiveness and superiority of our method in the weakly-supervised face forgery localization task, which mainly benefits from the novel localization approach based on iterative distribution estimation.

## 4.4 EXPERIMENTS ON FACEFORENSICS++

To assess the localization and detection ability to previous forgeries, we conduct the experiments on most widely-used FaceForensics++ (FF++), which is a large scale dataset for face forgery detection. We utilize five different manipulation types: DeepFakes (DeepFakes, 2019), Face2Face (Thies et al., 2016), FaceSwap (FaceSwap, 2019), NeuralTextures (Thies et al., 2019) and FaceShifter (Li et al., 2019). Since original FF++ lacks pixel-level annotations, a recent work (Miao et al., 2023) present a reconstructed version of FF++ dataset, incorporating more rigorous and rational pixel-level annotations, namely P-FF++. Models are developed on overall train dataset in the high quality setting (FF++.C23), and then evaluate on each forgery subset.

The results in Table 2 show that the proposed method consistently achieves superior localization and detection performance compared to other weakly-supervised methods. Comparing with the results of UIA-ViT (Zhuang et al., 2022), we find that the performance of our method is higher in localization performance. This is due to the different distribution estimation method in the weakly-supervised localization module. As for UIA-ViT, it adopts fixed region sampling for distribution estimation, and the asigned **GFR** more conforms to the actual manipulated regions of FF++. In spite of this, our method still achieves higher performance on the average localization and detection results than UIA-ViT, under the condition of assigning a rough initialization region for the localization module. It further demonstrates that our method is more adaptive to different manipulation types, which can dynamically conform to different manipulation regions with progressive update of distributions.

To intuitively illustrate the weakly supervised localization capacity, we visualize the **Predicted Location Map** of different datasets. In Fig.5, the model is developed on DiffFMD training datasets(PaintbyExample, Realistic, SDXL). The predicted location map is consistent with the ground-truth location map, which indicates the model can effectively locate the forgery regions.

## 4.5 ABLATION STUDY

To explore the effectiveness of different components of the proposed method, we spilt each part separately for verification. 'Fixed Sampling' represents the MVG estimation method in UIA-ViT (Zhuang et al., 2022). Compared to GradCAM and Fixed Sampling, our proposed MVG-FL with adaptive sampling strategy achieves the superior forgery localization ability. The DCL module also shows effectiveness in improving the forgery detection and localization performance.

The slack variable is another key component of adative sampling strategy. It determines the decision threshold and the sampling amount for the fake distribution in the current iteration stage. We compare the adaptive update approach (such as Eq.2 and Eq.3) with simply assigning the threshold as zero. The results show that adaptively updating the slack variable can achieve the superior localization

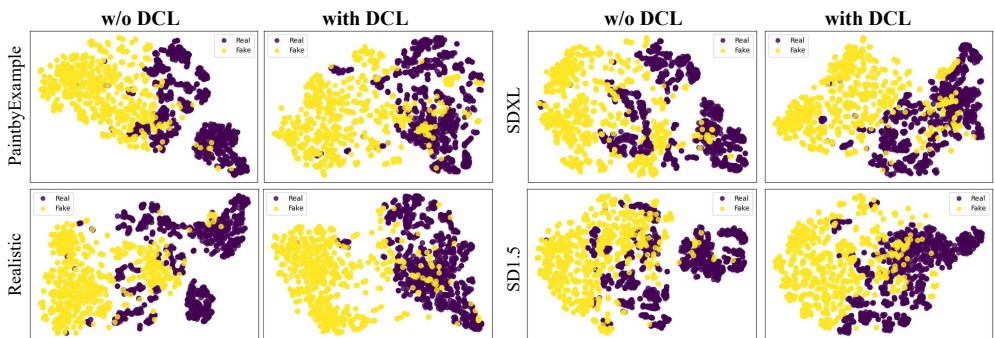

Figure 6: T-SNE visualization of model without DCL and with DCL module on different face manipulation testsets. The features are patch embeddings extracted from the 6-th layer of backbone, which are fed into the MVG-FL module for forgery localization.

Table 3: Abalation Study for the effect of different components

| GradCAM | MVG-FL | | | DCL | DiffFMD Test Set | | |
|---|---|---|---|---|---|---|---|
| | Fixed Sampling | Adaptive Sampling ($\hat{\theta}_{maha} = 0$) | Adaptive Sampling (update $\hat{\theta}_{maha}$) | | pix-AUC | pix-F1 | img-AUC |
| ✓ | | | | | 68.59 | 29.57 | 80.88 |
| | ✓ | | | | 85.63 | 31.37 | 81.08 |
| | | ✓ | | | 85.60 | 40.68 | 81.61 |
| | | | ✓ | | 86.48 | 46.90 | 81.74 |
| | | | ✓ | ✓ | **88.85** | **48.15** | **82.45** |

performance than the fixed threshold. It can flexibly regulate the decision threshold of relative distances, by controling fault-tolerant ability towards real samples.

Furthermore, to intuitively analyze the effectiveness of DCL module, we employ t-SNE visualization to compare the latent space distributions of the proposed model without/with the DCL module. As illustrated in Figure 6, the patch features of forged regions and real regions are extracted from the 6-th layer of the backbone. The latent representations learned with the DCL module are more compact compared to the other, and the distinction between features of real regions and forged regions is more pronounced. This observation suggests that our framework effectively learns more discriminative features that capture the underlying characteristics common to various manipulation models.

## 5 CONCLUSION

In this work, we present a novel weakly-supervised approach for face forgery localization based on iterative MVG distributions. The forgery localization module equipped with the adaptive sampling strategy is capable of handling multiple sizes of manipulations. Furthermore, the proposed distribution centrality learning strategy reinforces the localization process by improving the compactness of patch embeddings. The development of DiffFMD dataset further supports the evaluation in realistic scenarios, showing its effectiveness in localizing face forgeries in the weakly-supervised setting.

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

# A DATASET CONSTRUCTION

We first introduce the data preparation process, including the collection of source images and the generation of manipulation regions. We then describe the diffusion-based inpainting generators used in DiffFMD dataset.

## A.1 DATA PREPARATION

DiffFMD dataset is composed based on three real face image datasets, FFHQ (Kazemi & Sullivan, 2014), CelebA (Liu et al., 2015), Youtube. As for the Youtube source dataset, it is collected from the real videos of several deepfake datasets, including FaceForensics++ (Rossler et al., 2019), CelebDF (Li et al., 2020b) and FFIW(Zhou et al., 2021). Randomly sample three to five frames in each video, and extract cropped faces and filter out those of less $300 \times 300$ resolutions. We further sample partial images from FFHQ and CelebA, and then finally compose the source face set in the amount of 69k. For each face image of the dataset, we first use the face detection toolkit known as Dlib to meticulously extract the facial landmarks. This is followed by a confirmation of the exact locations of several key facial regions, namely the eyes, nose, mouth, upper-half face, bottom-half face and the entire face. To mimic the realistic face manipulation, we have pre-defined three distinct size levels of manipulation regions, which are categorized as tiny, small and big. The tiny region is specifically assigned as one of the eyes, nose, or mouth regions. The small region is the random combination of eyes, nose, and mouth, choosing two or three from these three facial regions. Lastly, the big region is one of the upper half face, the bottom half face, or the entire face. We finally conduct dilation and erosion operations on the randomly chosen region and generate the manipulation mask for diffusion-based inpainting generators.

## A.2 DIFFUSION-BASED IMAGE INPAINTING

There are diffusion-based inpainting generators we use for generating manipulated face images. We use SOTA diffusion-based inpainting generators for generating manipulated face imgs: PaintbyExample (Yang et al., 2023), Repaint (Lugmayr et al., 2022), Stable Diffusion (Rombach et al., 2022) (SD1.5, SD2 and SDXL), and Realistic Vision.

**PaintbyExample** (Yang et al., 2023) is an exemplar-guided image editing approach that allows accurate semantic manipulation according to an exemplar image. We use the pre-trained model[1] from HuggingFace library.

**Stable Diffusion** (Rombach et al., 2022) is a generative model that utilizes latent diffusion processes to generate high-quality images. We use three versions of stable diffusion for generating manipulated images, that are SD1.5-inpainting[2], SD2-inpainting[3], SDXL-inpainting[4].

**Repaint** (Lugmayr et al., 2022) produces faces and general-purpose image inpainting based on pre-trained unconditional DDPM. We use the pre-trained model[5] and generative schedule "RePaint-Pipeline" from diffusers.

**Realistic Vision** can generate visually convincing images that are difficult to distinguish from real photographs. It is developed by CivitAI [6] based on SD1.5 and we use the pre-trained inpainting model of Realistic Vision V5.0[7].

The detailed introduction about the composition of DiffFMD dataset is shown in Table 4. We choose three mainstream manipulation methods, namely PaintbyExample, Realistic and SDXL, to generate a large amount of train and valid sets, composed of 272k images. For each source image, we first randomly select a manipulation region from the three categories, and produce a mask image $M$ as input for the generator. The manipulated image $I_{diff}$ from the generator is

---

[1]https://huggingface.co/Fantasy-Studio/Paint-by-Example

[2]https://huggingface.co/diffusers/stable-diffusion-1.5-inpainting

[3]https://huggingface.co/stabilityai/stable-diffusion-2-inpainting

[4]https://huggingface.co/diffusers/stable-diffusion-xl-1.0-inpainting-0.1

[5]https://huggingface.co/google/ddpm-ema-celebahq-256

[6]https://civitai.com/

[7]https://civitai.com/models/4201?modelVersionId=125437

Table 4: The composition of DiffFMD dataset. Three random mask regions are generated for each source image. Then several diffusion-based generators are utilized for generating face manipulation results.

| DiffFMD | Category | Type | Number |
|---|---|---|---|
| Train & Valid Set | source | FFHQ CelebA Youtube | 68k |
| | local manipulation | PaintbyExample | 68k |
| | | Realistic | 68k |
| | | SDXL | 68k |
| | **Total** | | **272k** |
| Test Set | source | FFHQ CelebA Youtube | 1k |
| | local manipulation | PaintbyExample | 1k |
| | | Realistic | 1k |
| | | SDXL | 1k |
| | | SD1.5 | 1k |
| | | SD2 | 1k |
| | | Repaint | 1k |
| | **Total** | | **7k** |

then fused with the authentic image $I_{input}$ according to the mask $M$ of edited region: $I_{output} = M \times I_{diff} + (1 - M) \times I_{input}$. The test set contains manipulated images of six generators (PaintbyExample, Realistic, SDXL, SD1.5, SD2, Repaint), which is composed of 7k images.

## B EXPERIMENTS

### B.1 EXPERIMENT IMPLEMENTATION DETAILS

We adopt the ViT-Base architecture (Dosovitskiy, 2020) as backbone where the input patch size is $16 \times 16$ and the number of encoder layer is set to 12. In the training process, the model is optimized only by cross-entropy loss in the first interative stage, and optimized by the total loss in the next epoch. The batch size is set to 256 and the Adam optimizer with the initial learning rate 3e-4 is adopted. The learning rate is reduced when the validation accuracy arrives at plateau. The segmentation head is trained after 4 epoches and the total model is trained for 12 epoches. Our experiments are performed on three NVIDIA RTX A6000 GPUs.

### B.2 EVALUATION METRICS AND TASKS

Localization is the main task we tackle. We report pixel-level AUC (pix-AUC) and F1-score (pix-F1) as evaluation metrics for forgery localization, and use image-level AUC (img-AUC) for forgery detection. Note that the pix-AUC and pix-F1 metrics are calculated on fake images of the test data sets. The predicted forgery location results are produced by both the segmentation head and the MVG-FL module. To generate the final forgery mask, we perform a voting strategy: take the maximum score between the segmentation mask and the pseudo mask from MVG-FL as the final score.

### B.3 PATCH CONSISTENCY LEARNING (PCL)

PCL is a intra-frame consistency learning strategy that aims to learn the inconsistency between different regions of the fake image, proposed in UIA-ViT(Zhuang et al., 2022).

Define the mean Attention Map between different patch embeddings from $N - n$ to $N$ Transformer layers as $\Upsilon^P \in \mathbb{R}^{P^2 \times P^2}$. $\Upsilon^P_{(i,j),(k,l)}$ represents the consistency between the embedding in position $(i, j)$ and other patch embedding in position $(k, l)$, and higher value means two positions have higher consistency. With the approximate forgery location map **M**, the PCL loss is used to supervise the

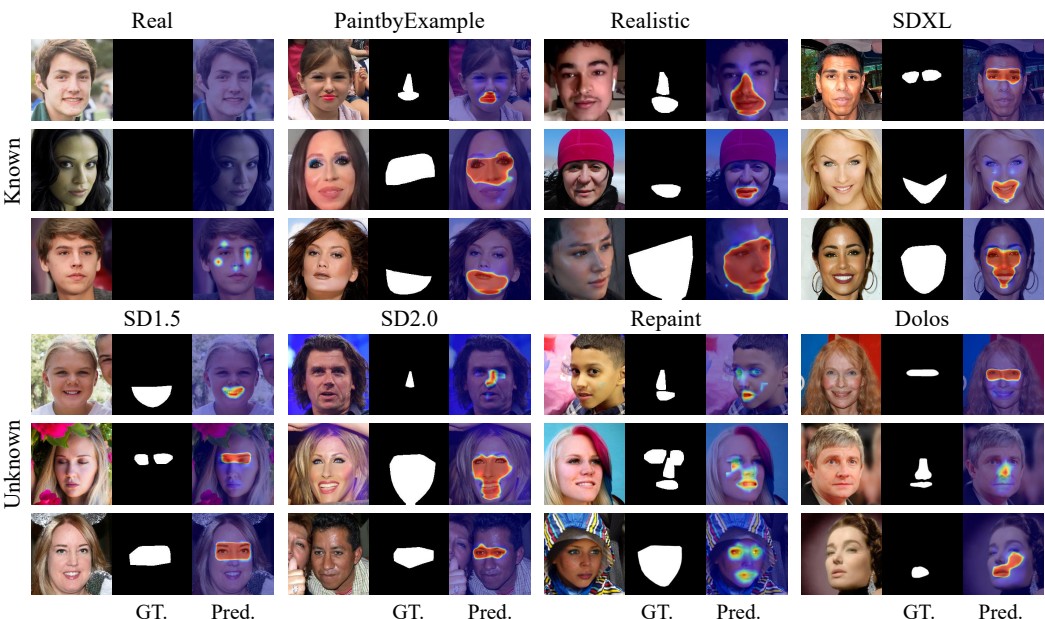

Figure 7: Predicted forgery location maps of DiffFMD test datasets.

Attention Map, formalized as:

$$\mathbf{C}_{(i,j),(k,l)} = \begin{cases} c_1, & if \quad \mathbf{M}_{ij} = 0 \quad and \quad \mathbf{M}_{kl} = 0 \\ c_1, & if \quad \mathbf{M}_{ij} = 1 \quad and \quad \mathbf{M}_{kl} = 1 \\ c_0, & else \end{cases}, \quad (9)$$

$$\mathcal{L}_{PCL} = \frac{1}{P^2} \sum_{i,j,k,l} \|\mathrm{sigmoid}(\Upsilon^P_{(i,j),(k,l)}) - \mathbf{C}_{(i,j),(k,l)}\|, \quad (10)$$

where $c_1$, $c_0$ are learnable parameters to avoid instability optimization when MVG estimation is biased in the early training. During training process, the initialize them as (0.6,0.2), and also optimize $c_1$ to increase and optimize $c_0$ to decrease gradually. After convergence, $c_1$, $c_0$ eventually tend to (1.0, 0.0) .

### B.4 VISUALIZATION

To intuitively illustrate the weakly supervised localization capacity, we visualize the predicted forgery location results of different datasets. In Fig.7, the model is developed on DiffFMD training datasets, consisting of three types of manipulation:ion: PaintbyExample, Realistic and SDXL. The ground-truth location map is shown in the middle column, and the predicted location map is shown in the right column. The predicted location map is consistent with the ground truth location map, which indicates that the model can effectively locate the forgery regions. The model can also predict the location of forgery regions in the unseen datasets, such as SD1.5, SD2.0, Repaint and Dolos, which are not included in the training datasets. It indicates the model can generalize to unseen manipulations. In Fig.8, the model is developed on FaceForensics++ training datasets, consisting of five types of manipulation: Deepfakes (DeepFakes, 2019), FaceSwap (FaceSwap, 2019), Face2Face (Thies et al., 2016), NeuralTextures (Thies et al., 2019) and FaceShifter (Li et al., 2019). We also find that most predicted forgery regions are consistent with the speculation.

### B.5 PARAMETER EXPERIMENT

In this section, we explore the effect of different parameters of the MVG-FL module.

Figure 8: Predicted forgery location maps of FaceForensics++ test datasets.

Table 5: Experimental results on DiffFMD datasets. Evaluate the forgery localization performance on pixel-level mIoU metrics.

| Methods | Test Datasets | | | | | | | | |
|---------|---------------|--------|------|-------|-------|---------|-------|-------|-------|
| | PaintbyExample | Realistic | SDXL | SD1.5 | SD2.0 | Repaint | Dolos | FF++ | Avg |
| GradCAM | 30.11 | 29.36 | 38.44 | 17.85 | 12.14 | 1.81 | 3.28 | 3.03 | 17.00 |
| Xception-Attn | 33.99 | 26.24 | 31.78 | 32.30 | 15.06 | 1.58 | 1.03 | 0.55 | 17.82 |
| Patchfor. | 24.93 | 24.96 | 25.17 | 19.12 | 14.71 | 14.16 | 16.19 | 27.09 | 20.79 |
| WSCL | 19.46 | 18.52 | 19.64 | 17.04 | 18.89 | 5.80 | 4.77 | 1.80 | 13.24 |
| POT | 28.63 | 28.46 | 26.19 | 29.04 | 20.74 | 3.32 | 7.95 | 6.28 | 18.83 |
| UIA | 17.65 | 17.12 | 17.31 | 14.13 | 14.45 | 15.43 | **24.28** | 40.19 | 20.07 |
| **Ours** | **34.28** | **31.44** | **33.76** | **34.33** | **34.16** | **29.67** | 23.34 | **47.74** | **33.59** |

Table 6: The effect of different parameters of Adaptive Sampling.

| top K | pix-AUC | pix-F1 | img-AUC |
|-------|---------|--------|---------|
| 48 | 88.43 | 47.94 | 81.94 |
| **64** | 88.85 | **48.15** | **82.45** |
| 96 | **88.97** | 48.10 | 81.51 |
| EMA parameter | pix-AUC | pix-F1 | img-AUC |
| 0.9 | 88.76 | 46.67 | 81.64 |
| **0.95** | 88.85 | **48.15** | **82.45** |
| 0.99 | 88.95 | 47.39 | 82.02 |

### B.5.1 PARAMETERS OF ADAPTIVE SAMPLING

We explore the proper hyper-parameters for adaptive sampling: K in Eq.2 and the EMA parameter $\alpha$ in Eq.3. The experimental results are shown in Table 6. We observe that the performance is less sensitive to hyper-parameters. Finally, we adopt K and $\alpha$ as 64 and 0.95, and achieve the best localization and detection performance.

### B.5.2 COMPARISON OF FIX REGION SAMPLING AND ADAPTIVE SAMPLING

As shown in Figure 10, the previous work UIA-ViT(Zhuang et al., 2022) conducts the MVG estimation through the fixed region sampling strategy, which assigns a fixed square region as the Genral Forgery Region (**GFR**) and samples patch embeddings within GFR for fake distribution estimation. The center square region is obviously limited and inflexible. Our method proposes an adaptive sampling strategy,

Table 7: Quantitative comparison experiments about Fixed Region Sampling and Adaptive Sampling.

| MVG Update | Initialization Region | pix-AUC | pix-F1 | img-AUC |
|---|---|---|---|---|
| Fixed Sampling (UIA) | Center Square | 85.63 | 31.37 | 81.08 |
| **Adaptive Sampling** | Full | 87.45 | 46.62 | 80.98 |
| | Center Square | 87.88 | 47.63 | 81.98 |
| | **Face Landmark** | **88.85** | **48.15** | **82.45** |

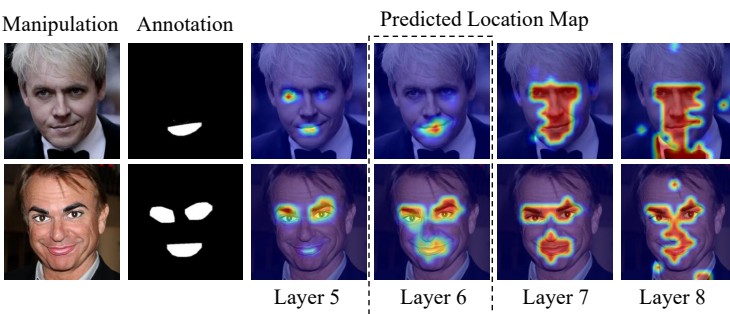

Figure 9: Predicted Location Map of different layers.

which can adjust the sampling region along with iterative update. Here we further make experiments to compare the performance of two strategies, and explore the effect of different initialization regions.

In Table 7, we show the performance of models equipped with two different sampling strategies. Three different initialization regions are assigned for adaptive sampling: (i) **Full** means adopting all patch embeddings within the fake samples as initial observations. (ii) **Center Square** means using patch embeddings within the same square region, like Fixed Region Sampling, as initial observations. (iii) **Face Landmark** means using patch embeddings within the region of facial landmarks (specifically eyes, nose and mouth). The results demonstrate that the adaptive sampling strategy outperforms the fixed region sampling strategy, especially for the forgery localization capacity. Additionally, dif-

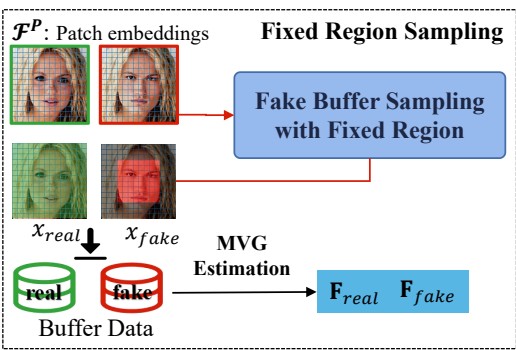

Figure 10: The pipeline of the sampling strategy (Fixed Region Sampling) of UIA-ViT.

ferent initialization regions also affect the efficacy of adaptive sampling. Since our dataset all consists of face images and most face forgery generators perform the manipulation on the facial region, it is rational to leverage the region of face landmarks for distribution initialization. The experimental results also confirm this procedure.

### B.5.3 DETERMINE WHICH LAYER FOR MVG ESTIMATION

In order to determine which layer is the best choice to conduct MVG estimation for forgery location prediction, we train the baseline ViT-Base model, and extract the patch embeddings from different middle layers for updating corresponding MVG distributions. Then, we visualize the predicted forgery location maps estimated by MVG distributions from 5-th, 6-th, 7-th and 8-th layers. Notice that different from the binary operation in Eq.6, here predicted location map is computed as $M_{ij} = \text{ReLU}(d(f_{ij}, F_{real}) - d(f_{ij}, F_{fake}))$, where non-zero value indicates predicting as fake.

The visualizations are shown in Fig.9. We observe that: (i) the distances between real and fake MVG distributions are larger in the later layers; (ii) the extracted features in the foreground and background

are also more distinguishable in the later layers; (iii) the predicted location map gradually expands to the whole image, because features in later layers capture more high-level semantic information rather than local texture information. Among them, we find that the predicted location map from 6-th layer is more closed to the annotation and apply it to conduct MVG estimation for forgery location prediction.

### B.6    LIMITATIONS

Our method is designed to address the weakly-supervised forgery localization task, and mainly explored effectiveness on our proposed DiffFMD dataset and FF++ dataset. Further research is needed to validate our performance on other SOTA forgery types.

### B.7    ETHIC IMPACTS

The proposed method strives to minimize negative social impacts and provide stronger protection for privacy, including the societal harms introduced by AIGC and the implications and challenges posed by detection errors. Specifically, our method improves the effectiveness and generalization ability of the deepfake detector, which could be crucial for reducing errors and mitigating the negative impact of malicious deepfake content.

