# OpenReview forum: "Weakly Supervised Forgery Localization and Detection Method for Face Manipulations"
_ICLR.cc/2026/Conference — Submitted to ICLR 2026_

### Official Review · Reviewer_WyMB · 2025-10-29

**Soundness:** 2
**Presentation:** 2
**Contribution:** 2
**Rating:** 4
**Confidence:** 4

**Summary:**

This paper presents a weakly supervised forgery localization framework that adaptively models feature distributions using multivariate Gaussian representations and introduces a distribution centrality learning scheme. In addition, the authors construct a new dataset, DiffFMD, to advance research in face forgery localization. Experimental results demonstrate that the proposed method achieves strong detection and localization performance on both the DiffFMD and FF++ datasets.

**Strengths:**

- This work proposes a MVG-based weakly supervised forgery localization framework that integrates MVG distribution estimation, adaptive sampling, and distribution centrality learning. The proposed adaptive learning and MVG-based distribution estimation schemes are well designed and conceptually interesting, and the experimental results are encouraging.

- The authors also construct a new face forgery dataset, DiffFMD, containing 279k images generated by several diffusion-based models, which will be valuable for advancing future research on face forgery localization.

**Weaknesses:**

- The motivation for the task of face forgery localization should be further clarified. What are the practical applications of face forgery localization? While I understand the rationale behind weak supervision in semantic segmentation due to the high cost and time of pixel-level annotation, the situation here is different. Since ground-truth masks of forged faces can be readily obtained during the fake generation process, the need for weak supervision should be explicitly justified to enhance clarity.

- Although MVG estimation is widely used in weakly supervised learning, it remains unclear why this particular statistical model is suitable for the task of face forgery localization. A more intuitive explanation or theoretical rationale would strengthen the paper.

- The details of L_PCL are suggested to be moved to the appendix for improved readability and flow of the main text.

- The proposed method does not compare against the SOTA techniques. The most recent baseline reported is from 2023. While this may be the first work on weakly supervised face forgery localization, the authors should at least include comparisons with contemporary weakly supervised semantic segmentation methods to contextualize performance.

- The authors use layer L for iterative distribution estimation and forgery localization, but the rationale behind this specific layer selection is not well explained. Clarifying this design choice would help readers understand the effectiveness of the approach.

- The contribution of the proposed dataset is somewhat limited. The dataset only incorporates diffusion-based forgery techniques. The authors should clarify how they ensure that the pixel values of real regions remain unchanged when using diffusion models to generate manipulated faces.

- The paper contains multiple typos and grammatical errors (e.g., “Sec. 4.1 Eq.equation 2, Eq.equation 3”). These should be carefully corrected throughout the manuscript.

- The method employs pix-AUC, pix-F1, and img-AUC to evaluate model performance. It is also recommended to include pix-IoU, which would offer a more intuitive measure for assessing localization accuracy.

**Questions:**

- The work lacks task-specific insight and design considerations for Deepfake detection and localization. The proposed framework appears to be an extension from natural image manipulation detection rather than one specifically tailored for facial forgery localization.

- It may be beneficial to explore introducing a new margin term in Eq. (5) to potentially further enhance the model’s discriminative capability.

---

> ### Author Response · Authors · 2025-12-03
>
> **Comment 1:**  Clarifying the motivation and practical applications of weakly supervised forgery localization.
>
> **Response:** We appreciate this feedback. Weak supervision remains crucial for two reasons:
>
> （1）Although masks can be obtained during synthesis, many real-world forgery datasets or in-the-wild forgeries **do not provide pixel-level masks**. Weak supervision allows localization using only image-level labels (real/fake), which is more scalable and applicable to scenarios where precise masks are unavailable.
>
> (2) Rapid Adaptation to New Forgery Techniques: Forgery techniques are rapidly evolving (e.g., new Deepfake models, smarter editing tools). When new types of forgeries emerge, collecting pixel-level annotated data for them lags significantly behind. In contrast, obtaining image-level labels for new forgery samples (e.g., knowing that a batch of newly generated images are fake) is much easier. Weakly supervised models can incorporate this new information much faster for adaptation and updates.
> As for practical applications, it includes forensic analysis, content moderation, and explainable detection, where pinpointing forged regions adds interpretability and trust.
>
> **Comment 2:** Explaining the reason why the MVG model is suitable for forgery localization.
>
> **Response:** The use of Multivariate Gaussian (MVG) modeling is motivated by both its established efficacy in related fields and its natural suitability for our specific task.
>
> First, in computer vision, modeling deep feature distributions with Gaussian assumptions is a classic and well-validated approach, especially in anomaly detection and weakly-supervised segmentation, where it helps characterize normative feature patterns and identify deviations.
>
> More importantly, for the task of forgery localization—which is inherently a per-pixel binary classification problem—the MVG provides a simple yet effective probabilistic framework to cluster features into “real” and “fake” categories. Under the Gaussian assumption, we can estimate the likelihood of each local feature belonging to the forgery distribution, which aligns well with our goal of separating authentic and manipulated regions in feature space.
>
> **Comment 3:** Add the details of $L_{PCL}$ to the appendix.
>
> **Response:** We appreciate the reviewer's suggestion. We added the detailed formulation of $L_{PCL}$ in the appendix.
>
> **Comment 4:** Supplementary comparison with the latest SOTA of weakly supervised semantic segmentation.
>
> **Response:** We appreciate for pointing out the deficiency in the comparison experiment. We supplement the comparison with POT[1], the latest SOTA methods of weakly supervised semantic segmentation published in 2025. The results are included in Table 1 and Table 2.
>
> [1] POT: Prototypical Optimal Transport for Weakly Supervised Semantic Segmentation. CVPR 2025
>
> **Comment 5:** Explaining the rationality of L layer selection.
>
> **Response:** Layer 𝐿 is chosen because it provides a balance between high-level semantic information and spatial detail. We conducted an visualization comparison on layer selection, showing that middle-to-deep transformer layers yield the best trade-off. Earlier layers lack semantics,  later layers capture more high-level semantic information rather than local texture information. Among them, we find that the predicted location map from 6-th layer is more closed to the annotation and apply it to conduct MVG estimation for forgery location prediction. We clarify this in Section B.3.3  of appendix.
>
> **Comment 6:** Emphasizing the contribution of the dataset and explaining the blending technology.
>
> **Response:**
> The contribution of dataset: **DiffFMD** is the first weakly supervised face forgery localization dataset focusing on diffusion model forgeries, filling the gap in the current research on diffusion forgery localization datasets. The dataset is designed to establish the reliable evaluation benchmark for weakly-supervised face forgery localization and detection task in a realistic and challenging scenario.
>
> The blending technology: While the diffusion model generates the entire image, our use of random masks strictly limits the generator’s conditioning to unmasked regions. Furthermore, we use the blending process after the diffusion generation. The manipulated image $I_{diff}$ from the generator is then fused with the authentic image $I_{input}$ according to the mask $M$ of edited region: $I_{output} = M \times I_{diff} + (1-M)\times I_{input}$. We clarify this details in the Section A.2 of the appendix.
>
> **Comment 7:** Correcting typos and formatting errors in the text.
>
> **Response:** We sincerely apologize for these errors. We thoroughly proofread the manuscript and correct all typos issues.
>
> **Comment 8:** Adding the pix-IoU evaluation metrics.
>
> **Response:** We add **pix-IoU** results to evaluate the forgery localization performance on DiffFMD in Table 5 of the appendix.

---

### Official Review · Reviewer_miqg · 2025-10-31

**Soundness:** 3
**Presentation:** 3
**Contribution:** 4
**Rating:** 6
**Confidence:** 3

**Summary:**

This paper addresses the task of localizing manipulated regions in forged facial images using only image-level labels (i.e., "real" or "fake", without extra pixel-level annotations). The authors propose a novel weakly-supervised framework, i.e., Multivariate Gaussian-based Forgery Localization (MVG-FL) and Distribution Centrality Learning (DCL) based on a ViT model. The authors also propose a a new large-scale dataset DiffFMD which generated by various state-of-the-art diffusion-based inpainting models.

**Strengths:**

1.This paper leverages dataset-wide statistical distributions (MVG) for localization instead of single-image features. The adaptive sampling strategy improves upon fixed sampling with added flexibility and theoretical support

2.The proposed  DiffFMD dataset fills a gap by focusing on diffusion-based inpainting manipulations of facial images, offering multiple manipulation sizes and generators as a valuable benchmark for real-world generalization

3.Extensive experiments on diverse test sets and thorough ablation studies clearly demonstrate the effectiveness of adaptive sampling and DCL

**Weaknesses:**

1.While adaptive sampling and DCL improve the method, the core MVG framework largely builds on UIA-ViT, making the contributions feel like incremental refinements rather than a major new direction

2.The method’s iterative distribution estimation, buffer updating, and multiple losses add complexity, but the paper lacks discussion on training time, resource use, or feasibility compared to simpler baselines

3.The ablation study shows only modest gains from DCL, and more detailed explanation or visualization of how DCL improves feature compactness is needed to clarify its value

**Questions:**

See weaknesses

---

> ### Author Response · Authors · 2025-12-03
>
> **Comment 1:** Emphasizing research contributions and distinguishing between incremental improvements and core innovations.
>
> **Response:** We appreciate this feedback. Our main contributions are: (1) introducing the novel task of **weakly-supervised face forgery localization**, which differs fundamentally from full supervision or detection; (2) proposing an **iterative MVG estimation framework** tailored for forgery localization, which dynamically updates feature statistics and refines localization maps; (3) designing **Distribution Centrality Learning (DCL)** to enhance feature compactness and further bolster the unsupervised forgery localization module; (4) The first weakly supervised localization dataset **DiffFMD** focusing on diffusion model forgeries was constructed, filling the gap in the current research on diffusion forgery localization datasets. While inspired by UIA-ViT’s MVG concept, our work adapts and extends it significantly for a new domain and task. We will better highlight these contributions in the introduction and conclusion.
>
> **Comment 2:** Discussion on training time and resource usage.
>
> **Response:**  We appreciate the reviewer's attention to this important practical issue. Compared to the baseline ViT model, our method introduces a moderate increase in training time due to the iterative MVG estimation and buffer updating processes. In our experimental setup (using 3×A6000 GPUs), training the baseline ViT required approximately 16 hours, while our full model took about 22 hours. The primary resource consideration is GPU memory. Since our MVG-FL framework maintains a feature buffer for iterative distribution estimation, peak memory usage during training reaches about 24 GB per epoch under our current configuration (with 4 fake and 2 real images are sampled and stored per batch).
>
> We note that this configuration is adjustable based on available hardware—for instance, by reducing the buffer size, sampling fewer examples per batch, or increasing the buffer update interval. These adjustments allow our method to remain flexible and feasible in practical deployment scenarios.
>
> **Comment 3:** Supplementary detailed explanation and visualization of DCL.
>
> **Response:** We appreciate the reviewer's suggestions for improvement. We enhance the analysis of DCL in Section 4.5 and Figure 6. Specifically, we include **t-SNE visualizations** of features with/without DCL to illustrate improved compactness and inter-class separation.

---

### Official Review · Reviewer_mmEW · 2025-11-01

**Soundness:** 4
**Presentation:** 3
**Contribution:** 3
**Rating:** 6
**Confidence:** 4

**Summary:**

This paper proposes a weakly supervised method for face forgery localization and detection based on a Vision Transformer (ViT) backbone. Unlike most prior works that rely on pixel-level annotations, the proposed MVG-FL approach leverages multivariate Gaussian (MVG) distribution estimation to model real and fake patch features, combined with an adaptive sampling strategy and Distribution Centrality Learning (DCL) to enhance embedding compactness. Furthermore, the authors construct a new dataset, DiffFMD, including manipulated images generated by multiple diffusion-based inpainting models with varying manipulation sizes. Extensive experiments demonstrate superior performance on both DiffFMD and FaceForensics++ datasets.

**Strengths:**

The paper proposes a weakly supervised localization framework based on multivariate Gaussian distributions, which is both novel and inspiring. It achieves superior performance in both forgery detection and localization tasks. Moreover, the paper introduces a newly constructed dataset, which can serve as a new standard benchmark for research on diffusion-model-based forgery detection.

**Weaknesses:**

1, In terms of the experimental setup, the results on the FF++ dataset appear to be obtained from a model trained on the full training set. However, it remains unclear whether a model trained on the DiffFMD dataset can directly transfer and generalize to the FF++ dataset. This aspect is crucial for evaluating the model’s practical applicability and robustness in real-world cross-domain forgery detection scenarios. It is recommended that the authors provide further clarification or additional experimental evidence on this point.
2, The paper does not appear to specify the release plan or privacy compliance of the DiffFMD dataset, which may affect the credibility of the work. Is there any plan to make the DiffFMD dataset publicly available?

**Questions:**

Refer to Weaknesses

---

> ### Author Response · Authors · 2025-12-03
>
> **Comment 1:** Explanation of the transfer generalization ability of the model trained on DiffFMD to the FF++ dataset.
>
> **Response:** Thank you for raising this important point. To evaluate cross-domain generalization, we conducted additional experiments where the model was trained solely on our **DiffFMD** dataset and then tested directly on **FF++** without any fine-tuning. The results demonstrate competitive performance compared to other methods, confirming the strong transferability and robustness of our approach. We include these results in the revised manuscript (Table 1) with a dedicated discussion on cross-domain generalization.
>
> **Comment 2:** Public release plan and privacy compliance of the DiffFMD dataset.
>
> **Response:** We fully agree on the importance of dataset availability. We plan to release the **DiffFMD** dataset publicly upon acceptance of the paper. All collected face images are from publicly available sources with permission for research use (such as FFHQ, CelebA-HQ and the real subset of FF++). We will also provide detailed data usage guidelines and ensure compliance with relevant privacy regulations.

---

### Meta-Review · Area_Chair_UQua · 2025-12-31

**Summary:**

The paper proposed a weakly supervised deepfake localization method using only image-level labels, and it further used multivariate Gaussian (MVG) distribution estimation to model forgery. It also introduced a new deepfake dataset (DiffFMD) using several pretrained diffusion-based generators.

Reviewers are likely to appreciate the cross-domain experiments, which significantly strengthen the paper's empirical standing. The introduction of the DiffFMD dataset provides a clear community contribution.
However, there are still several unsolved concerns:
1) It is unclear whether the approach is a significant advancement over existing models like UIA-ViT.
2) The motivation for the task of face forgery localization should be further clarified.
3) The paper lacks a clearer discussion of training time, resource usage, and practical feasibility compared to simpler baselines.
Given that the paper's initial concerns slightly outweigh its strengths, I recommend rejection.

**Reviewer Concerns:**

- Reviewer mmEW's concerns have been properly addressed.

- Reviewer miqg's concern regarding the technical novelty might not be well solved. While the authors highlighted their unique task (weakly supervised face forgery), the technical overlap with the UIA-ViT framework is still open to interpretation and may remain unconvincing to the reviewer. Moreover, the reviewer requested a clearer discussion of training time, resource usage, and practical feasibility compared to simpler baselines. The rebuttal only provides a limited training-time comparison, and only against a single baseline (ViT), which may not be sufficient to resolve these concerns.

- Reviewer Z3cx didn't finish the review, but reported that the paper contains blank markers which may violate the submission requirements. I have checked them. They are all figure titles, which I believe were generated by mistake.

- Reviewer WyMB: While the authors provided an intuitive explanation for using MVG, some reviewers may still look for a more rigorous mathematical proof or unique "face-specific" inductive biases beyond general anomaly detection logic.

**Reviewer Scores:**

- Reviewer mmEW: keep score 6
- Reviewer miqg: keep score 6 or downgrade to 4
- Reviewer Z3cx: N/A
- Reviewer WyMB: 4

---

### Decision · Program_Chairs · 2026-01-26

Reject